# One-Model-Connects-All: A Unified Graph Pre-Training Model for Online Community Modeling

**Ruoxue Ma[1], Jiarong Xu[2]\*, Xinnong Zhang[1], Haozhe Zhang[5], Zuyu Zhao[5],**
**Qi Zhang[3], Xuanjing Huang[3], Zhongyu Wei[1, 4]\***

[1]School of Data Science, Fudan University, China
[2]School of Management, Fudan University, China
[3]School of Computer Science, Fudan University, China
[4]Research Institute of Intelligent Complex Systems, Fudan University, China
[5]Huawei Technologies Co.,Ltd, China
{rxma21, xnzhang23}@m.fudan.edu.cn
{jiarongxu, qz, xjhuang, zywei}@fudan.edu.cn
{zhanghaozhe7, zhaozuyu1}@huawei.com

## Abstract

Online community is composed of communities, users, and user-generated textual content, with rich information that can help us solve social problems. Previous research hasn't fully utilized these three components and the relationship among them. What's more, they can't adapt to a wide range of downstream tasks. To solve these problems, we focus on a framework that simultaneously considers communities, users, and texts. And it can easily connect with a variety of downstream tasks related to social media. Specifically, we use a ternary heterogeneous graph to model online communities. Text reconstruction and edge generation are used to learn structural and semantic knowledge among communities, users, and texts. By leveraging this pre-trained model, we achieve promising results across multiple downstream tasks, such as violation detection, sentiment analysis, and community recommendation. Our exploration will improve online community modeling.

## 1 Introduction

An online community refers to a group of individuals who share similar interests or purposes and utilize the internet as a means to communicate with one another (Haythornthwaite, 2007; Baym, 2000). It serves as a vital component of social networks, including platforms like NextDoor and Facebook groups. Typically, an online community is composed of *users*, *user-generated textual content*, and *communities*. Members within the same community often share common languages, behavior rules, and social cognition, and actively contribute to the community by posting or commenting on various

---

\*corresponding author

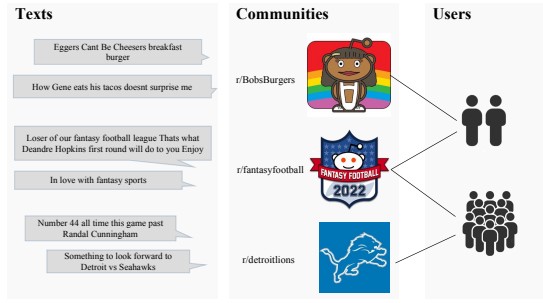

Figure 1: An illustration of the connection among communities, users, and texts. Take Reddit, which has many online communities called subreddit, as an example. Similar subreddit r/fantasyfootball and r/detroitlions share more common users and have more similar semantics, while r/fantasyfootball and r/BobsBurgers do not. We slightly process the text for privacy protection.

topics and discussions (Haythornthwaite, 2007; Baym, 2000).

There are many studies focused on integrating online community to solve social problems such as abusement detection (Park et al., 2021), conflict prediction (Kumar et al., 2018), and bias study (Strzalkowski et al., 2020). Previous research mainly aims at a specific social problem. Although they achieve positive results on target tasks, their methods require designing different models for different tasks, which is very costly. The training process also requires high-quality, annotated datasets, which is very time-consuming and labor-intensive. In addition, there is a strong connection among communities, users, and texts as shown in Figure 1. But previous research did not take into account all these three components simultaneously. Some work only modeled the relationship between communities and users (Kumar et al., 2018; Martin, 2017; Waller and Anderson, 2019), while others

only utilize the community and their textual content (Strzalkowski et al., 2020; Trujillo et al., 2021). What's more, these studies model the online communities' relationships in a simple way such as appending community information to the text as an additional contextual input (Park et al., 2021).

In this paper, we target a unified pre-training framework that simultaneously models community, textual content, and user along with their information flow, such that one pre-trained model can support a wide range of downstream tasks for online community.

First, to achieve this, we begin by constructing a heterogeneous graph that captures the complex relationships among communities, users, and textual content. The constructed graph consists of three types of nodes (*community*, *user*, and *text*) and three types of edges (*post in*, *post by* and *active*). This facilitates the subsequent pre-training tasks can capture community-level, user-level, and text-level information and the connection among different levels.

Next, we introduce two self-supervised tasks such that the pre-trained model built upon the constructed heterogeneous graph can encode rich knowledge of communities, users, and texts. Text reconstruction task is designed to enable the model to learn the language style of communities and users, while the edge generation task aims to focus on the connection among texts, communities, and users.

Third, to support different downstream tasks within online community, we fine-tune the pre-trained model on various downstream tasks such as text classification and community recommendation. By simply adding a fully connected layer, our model can be easily applied to community-level, user-level, and text-level tasks. Extensive experiments have been completed to confirm the effectiveness of our framework.

Our main contributions are as follows: Firstly, we propose a unified framework containing communities, users, and textual content simultaneously. Secondly, we employ graph-based approaches to model the relationship among these three elements. Lastly, we use two self-supervised tasks to pre-train this ternary graph, making it easy to adapt to multiple downstream tasks related to social media.

## 2 Proposed Framework

The proposed framework, as illustrated in Figure 2, consists of three components: Graph Construction, Pre-training, and Downstream Tasks.

### 2.1 Graph Construction

#### 2.1.1 Data Collection

Reddit[1] is a social media website that is divided into different online communities called subreddit. Users can post in topic-specific subreddit and comment posts in their interested subreddit. Subreddit inherently possesses the characteristics of online community, which makes Reddit an ideal data source for studying community profiling.

In 2021, the total number of subreddits has exceeded 100,000. Although there are a large number of subreddits, we find that only 2% of subreddits can cover 75% of the posts. So we only use a subreddit collection, which consists of the top 2% of subreddits by number of posts per month, as our research object.

There are two types of text in Reddit: *submissions* and *comments*. *Submissions* are posts in subreddit and *comments* are comments. *Comments* are threaded: *Comments* can be made directly on *submissions* or appear as a reply to another comment. After deleting empty texts, deleted texts, removed texts, texts published by deleted users, too long or too short text, and all signed text, we sample 1.5% of *submissions* and 1‰ of *comments* as research subjects. The small sampling proportion of comments is due to the vast quantity of comments, coupled with their lower semantic meaning compared to submissions.

We define high-frequency users as those whose number of submissions and comments falls in the top 25% of the subreddit annually. Due to the huge amount of data, we only focus on the high-frequency users among the authors of above sampled texts. Table 3 in A.1 presents the basic summary statistics of our dataset.

#### 2.1.2 Heterogeneous Graph Construction

Heterogeneous graph consists of three types of nodes.

- $C = \{c_1, c_2, ..., c_{N_c}\}$ is the set of community nodes, where $c_i(i = 1, 2, ...)$ is the i-th community node and $N_c$ is the number of community nodes.

---

[1]https://www.reddit.com

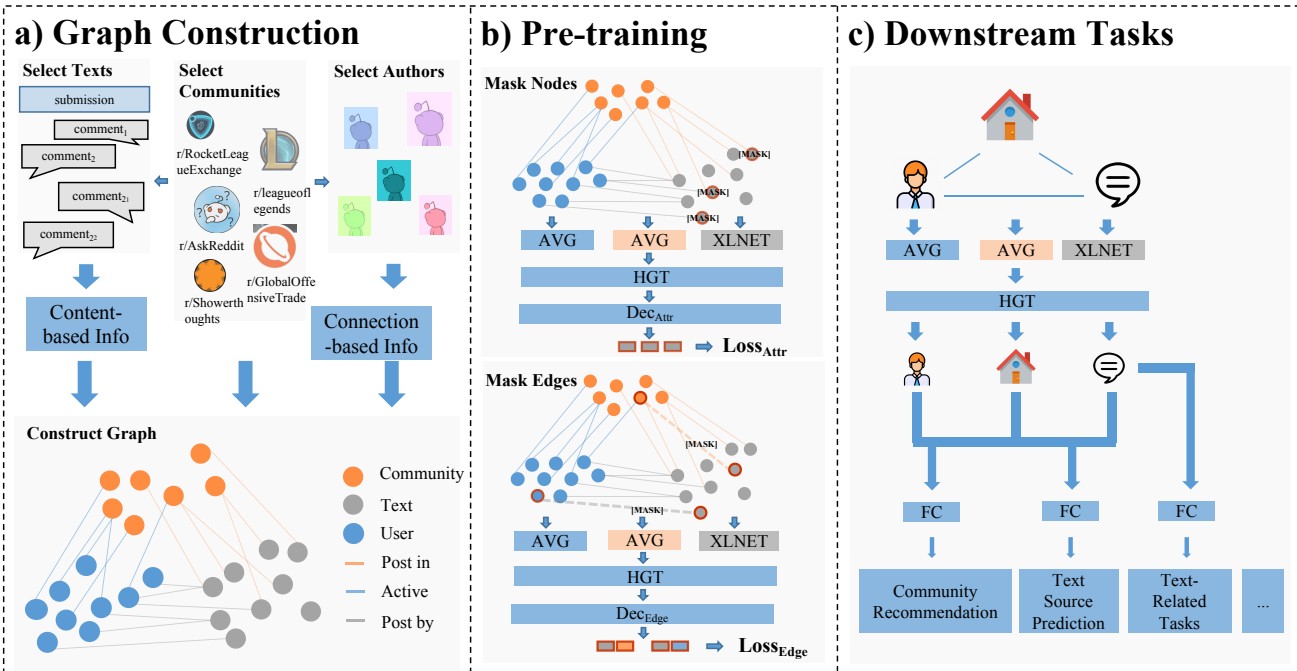

Figure 2: The proposed framework architecture. a) Graph Construction. We collect data and construct a ternary heterogeneous graph incorporating community, user, and text nodes. b) Pre-training. Two self-supervised tasks are used to effectively capture and leverage the rich knowledge of the graph. c) Downstream Tasks. We transfer the learned parameters to downstream tasks and add fully connected layers to adapt to different objectives.

- $T = \{t_1, t_2, ...t_{N_t}\}$ is the set of text (submission and comment) nodes, where each $t_i(i = 1, 2, ...)$ is the i-th text and $N_t$ is the number of text nodes.

- $U = \{u_1, u_2, ...u_{N_u}\}$ is the set of users, where each $u_i(i = 1, 2, ...)$ is the set of user nodes and $N_u$ is the number of user nodes.

Three types of edges can be constructed by connecting three types of nodes.

- Relations between communities and texts (post in). In Reddit, each text is posted in corresponding community, so it is easy to get the relationship between text nodes and community nodes. By connecting the community node and text node, text node can obtain supplementary information from their community and community node can aggregate the information from all texts posted in it to get better representation. We construct an adjacency matrix $A$. Element $a_{ij}$ indicate whether the community $c_i$ contains the text $t_j$.

- Relations between communities and users (active). We define high-frequency users as those whose number of submissions and comments

is in the top 25% of the subreddit annually. By connecting the community node and user node, user node can obtain supplementary information from their highly engaged communities and community node can aggregate the information from their high-frequent users.

- Relations between texts and users (post by). This type of edge represents the affiliation between the text and its author. We construct an adjacency matrix $B$. Each element $b_{ij}$ indicates whether the user $u_i$ writes this text $t_j$.

## 2.2 Pre-training

### 2.2.1 Initialization of Nodes

Large-scale pre-trained language models have learned rich semantics and can provide strong support for various natural language processing tasks (Devlin et al., 2018; Liu et al., 2019; Zhang et al., 2020) . Due to the rich textual information of text nodes, we use XLNet (Yang et al., 2019) to initialize the text nodes.

$$X_{t_i} = XLNet(clean(t_i)) \qquad (1)$$

$clean$ is text preprocessing function, which includes removing symbols, modifying "hyperlinks"

and "mentioning users" to [URL] and [USER]. Other nodes are initialized by averaging the vectors of text nodes connected to them.

$$X_{c_i} = AVG(X_{t_j}), a_{ij} = 1 \qquad (2)$$

$$X_{u_i} = AVG(X_{t_j}), b_{ij} = 1 \qquad (3)$$

$AVG$ represents the average operation, and $t_j$ represents the text nodes connected to subreddit nodes and user nodes. After initializing, we obtain $X_c \in \mathbb{R}^{N_c \times d}$, $X_u \in \mathbb{R}^{N_u \times d}$, $X_t \in \mathbb{R}^{N_t \times d}$.

### 2.2.2 Graph Neural Network

We then feed the output of initializing into HGT (Hu et al., 2020b) module to update their representations. HGT is a graph neural network model which introduces attention mechanisms related to node and edge types to solve graph heterogeneity. It consists of three parts: Heterogeneous Mutual Attention, Heterogeneous Message Passing, and Target Specific Aggregation. After HGT module, we obtain: $h_c = HGT(X_c)$, $h_u = HGT(X_u)$, $h_t = HGT(X_t)$, where $h_c \in \mathbb{R}^{N_c \times d}$, $h_u \in \mathbb{R}^{N_u \times d}$, $h_t \in \mathbb{R}^{N_t \times d}$.

### 2.2.3 Pre-training Process

There are two self-supervised tasks in GPT-GNN (Hu et al., 2020a): node generation and edge generation. Inspired by GPT-GNN, we construct text reconstruction and edge generation on our dataset. Same as GPT-GNN, we need to separate the text nodes into *text reconstruction nodes* and *edge generation nodes* to prevent information leakage during edge generation.

**Text Reconstruction**. We mask the *text reconstruction nodes* with a dummy token and define the output of the dummy token generated by 2.2.2 as $h^{Attr}$. We also denote the decoder of attribute generation as $Dec_{Attr}()$. $Dec_{Attr}()$ takes $h^{Attr}$ as input and generates the predicted attributes. A distance function is used to calculate the distance between the generated attributes and the original attributes. This distance function is also the loss function of this task.

$$L^{Attr} = AVG(Distance(Dec_{Attr}(h_{t_i}^{Attr}), X_{t_i})),$$
$$i = 1, 2, ... \qquad (4)$$

Minimizing the distance function is equivalent to maximizing the likelihood of observing the text. In this way, the pre-trained model can capture the semantic of communities and users.

**Edge Generation**. Some edges are masked at this stage. After 2.2.2, the output of the i-th node can be obtained as $h_i^{Edge}$. We denote the set of nodes connected to the above i-th node but their edge is masked as $S_i^+$, $j+ \in S_i^+$. We also denote the decoder of edge generation as $Dec_{Edge}()$. For each $i$ and $j+$, we sample some unconnected nodes of the same type as $j+$ and denote this set as $S_i^-$. Then, contrastive loss can be calculated via

$$L_i^{Edge} = -\sum_{j+ \in S_i^+} \frac{exp(Dec_{Edge}(h_i^{Edge}, h_{j+}^{Edge}))}{\sum_{j \in S_i^- \cup \{j+\}} exp(Dec_{Edge}(h_i^{Edge}, h_j^{Edge}))} \qquad (5)$$

$$L^{Edge} = AVG(L_i^{Edge}), i = 1, 2, ... \qquad (6)$$

Optimizing $L^{Edge}$ is equivalent to maximizing the likelihood of generating edges. In this way, the pre-trained model can capture the intrinsic structures of graph, which means it can better learn the connections among users, texts, and communities.

**Overall Pre-training**. Final loss is linearly combined by above two factors:

$$L^{Total} = \lambda L^{Attr} + (1 - \lambda) L^{Edge} \qquad (7)$$

where $\lambda$ is a hyperparameter controlling the weight of two losses.

## 2.3 Downstream Tasks

### 2.3.1 Methodology

We select several Reddit-related datasets and build them into the same graph structure as the above process. We use pre-trained model parameters to initialize the model of downstream tasks and add a fully connected layer to adapt to different objectives. Our downstream tasks can be divided into node-related and edge-related.

### 2.3.2 Datasets

Experiments are conducted on five Reddit-related datasets. We perform additional crawls for text without community and author details. Preserving the structure of original datasets, we remove samples whose text has been deleted or whose author has been deleted. Therefore, the composition of the dataset differs from the original. We report the size and partitioning of each dataset in the appendix.

**Node-related Downstream Tasks**

| | | Node-related Tasks | | | | | | Edge-related Tasks | | | |
|---|---|---|---|---|---|---|---|---|---|---|---|
| | | NORMVIO | | Ruddit | | Dreaddit | | Anxiety-on-Reddit | | Community Recommendation | |
| | | ACC | F1 | r | MSE | ACC | F1 | ACC | F1 | NDCG | MRR |
| Weak Baselines | Majority | 0.611 | 0.379 | - | - | 0.516 | 0.340 | 0.530 | 0.346 | 0.139 | 0.025 |
| | Random | 0.486 | 0.480 | 0.014 | 0.439 | 0.492 | 0.492 | 0.503 | 0.502 | 0.101 | 0.003 |
| Neural Networks | LSTM | 0.737 | 0.702 | 0.825 | 0.036 | 0.767 | 0.763 | 0.779 | 0.773 | - | - |
| Pretrained Language Models | BERT | 0.754 | 0.720 | 0.854 | 0.027 | 0.791 | 0.790 | 0.808 | 0.806 | - | - |
| | XLNet | 0.758 | 0.736 | 0.853 | 0.030 | 0.785 | 0.784 | 0.814 | 0.808 | - | - |
| | Hate BERT | 0.754 | 0.739 | **0.865** | **0.026** | - | - | - | - | - | - |
| Graph Models | GAT | 0.758 | 0.731 | 0.838 | 0.032 | 0.784 | 0.784 | 0.796 | 0.790 | 0.184 | 0.076 |
| | HAN | 0.762 | 0.740 | 0.843 | 0.030 | 0.785 | 0.784 | 0.807 | 0.803 | 0.193 | 0.087 |
| | HGT | 0.760 | 0.744 | 0.841 | 0.031 | 0.793 | 0.791 | 0.800 | 0.798 | 0.196 | 0.087 |
| *ours* | *ours* | **0.773** | **0.755** | 0.854 | 0.027 | **0.809** | **0.807** | **0.826** | **0.825** | **0.205** | **0.095** |

Table 1: Comparison results of our models and baselines. (average of 5 runs). We report ACC(accuracy) and F1(macro-f1) for NORMVIO, Ruddit and Anxiety-on-Reddit, r(Pearson's R) and MSE(Mean Squared Error) for Ruddit, NDCG(Normalized Discounted Cumulative Gain) and MRR(Mean Reciprocal Rank) for Community Recommendation.

- NORMVIO (Park et al., 2021) is a new dataset focusing on a more complete spectrum of community norms and their violations. It includes Reddit comments which are removed for violating community norms. We obtain the original comments based on IDs provided by the author and reconstruct the task as a binary classification to determine whether it is a violation comment.

- Ruddit (Hada et al., 2021) is the first dataset of English language Reddit comments that has fine-grained, real-valued scores between -1 (maximally supportive) and 1 (maximally offensive). It is annotated with Best–Worst Scaling, a form of comparative annotation that has been shown to alleviate known biases of using rating scales.

- Dreaddit (Turcan and McKeown, 2019) is a Reddit dataset for stress analysis in social media. Stress is common in daily life, particularly in online world. Too much stress is related to physical and mental health (Lupien et al., 2009; Calcia et al., 2016). Therefore it is very important to recognize stress. Dreaddit consists of posts from five different categories of Reddit communities.

**Edge-related Downstream Tasks**

- Anxiety-on-Reddit (Shen and Rudzicz, 2017) is a dataset to study anxiety disorders through personal narratives collected through Reddit. In this dataset, anxiety posts come from several anxiety-related subreddits. Due to the unavailability of original data information, we rebuild a dataset similar to Anxiety-on-Reddit. Concretely, we crawl the data following the data collection process described in paper. The time period is from January 2018 to March 2018. The label of text is determined by the community to which they belong. Therefore, we model this task as a community-text edge prediction task. When making predictions, we block the text community edge to prevent label leakage.

- Recommend community for users. The sociological theory of *homophily* asserts that individuals are usually similar to their friends (McPherson et al., 2001). People are more likely to share with their friends rather than random individuals (Yang and Eisenstein, 2017). So we design a downstream task for recommending communities to users to help users find friends with similar interests or purposes. Concretely, we crawl the data in January 2018 and built them into graph as described in 2.1. Recommending communities task is modeled to predict which community will have an edge connection with the targeted

user. When making predictions, we block the text community edge to prevent label leakage.

# 3 Experiment

## 3.1 Data preprocessing

Our pre-training is built on Reddit data of the whole year 2017. After data sampling introduced in 2.1.1, we perform data preprocessing on the text nodes. We first remove the special symbols contained in text and then replace hyperlinks and user mentions with [URL] and [@USER]. Finally, we keep only the first 510 words for too long text.

There are too many nodes in this graph, so it is difficult to train the entire graph. We sample subgraphs for training by HGSampling algorithm (Hu et al., 2020b). The above sampling methods guarantee a high connectivity between sampled nodes and preserve maximal structural information. Following GPT-GNN's approach, Adaptive Queue (Hu et al., 2020a) is used to alleviate the problem that $L^{Edge}$ only has access to the sampled nodes in a subgraph.

## 3.2 Implementation Details

We set sample depth to 6 and sample width to 256 for sampling subgraphs. The head number and layer number of HGT are set as 8 and 3. During pre-training, we set the decoder $Dec_{Attr}()$ and $Dec_{Edge}()$ as multi-layer perception and distance function as cosine similarity. The size of adaptive queue is set as 256. The hidden dimension $d = 768$ (same as the output dimension of pre-trained language model) is set for all modules. We optimize the model via the AdamW optimizer (Loshchilov and Hutter, 2017) with the Cosine Annealing Learning Rate Scheduler (Loshchilov and Hutter, 2016). We pre-train the model with 5 epochs to get the final parameters of graph network. The codes are implemented by Pytorch and PyTorch Geometric(PyG). The entire pre-training procedure consumes a lot of memory, so it is conducted on only one GeForce RTX 3090 GPU.

## 3.3 Compared Models

**Weak Baselines.** Majority generates the label with the highest proportion and Random generates random labels. Majority is not used in Ruddit because the label in Ruddit is continuous.

**Neutral Models.** LSTM (Hochreiter and Schmidhuber, 1997) is a type of recurrent neural network which is designed to address the problems of long-term dependencies in sequential data.

**Pre-trained Language Models.** BERT (Devlin et al., 2018) and XLNet (Yang et al., 2019) are two pre-trained language models that leverage transformer-based architectures and are pre-trained in general corpus. HateBERT (Caselli et al., 2020) is a re-trained BERT model for abusive language detection in English. We only test HateBERT in NORMVIO and Ruddit because only these two datasets are related to abusive language. Baseline pre-trained language models are continued pre-training on the collected Reddit data. We don't test pre-trained language models in community recommendation task because this task is not related to texts.

**Graph-based Models.** GAT (Veličković et al., 2017), HAN (Wang et al., 2019) and HGT (Hu et al., 2020b) are three graph-based models which can capture structural information in data. XLNet is used to initialize the node representation similar to 2.2.1.

## 3.4 Experiment Result

**Node-related Tasks.** Our model shows very good performance on node-related tasks as shown in Table 1, which indicates combining the structure of social media can promote text comprehension. Our model performs worse than HateBERT on Ruddit. we think it is due to the small size of Ruddit and the lack of concentration of communities and authors, which means there are only a few texts coming from the same user or community.

**Edge-related Tasks.** Our model also performs well on edge-related tasks as shown in Table 1. This means our model can effectively learn the structural information contained in the graph during the pre-training stage.

# 4 Further Analysis

## 4.1 Ablation Study

We conduct ablation studies to explore the effects of different components. Results are reported in Table 2. We divide datasets into text-related and user-related, and we remove user information and text information respectively. We also analyze the impact of pre-training module. As shown in Table 2, structural information is useful to better model texts, and semantics can also assist in structural learning. Additionally, pre-training has been proven to be useful.

| | Text-related Tasks | | | | | | | | | User-related Tasks | |
|---|---|---|---|---|---|---|---|---|---|---|---|
| | NORMVIO | | Ruddit | | Dreaddit | | Anxiety-on-Reddit | | | Community Recommendation | |
| | ACC | F1 | r | MSE | ACC | F1 | ACC | F1 | | NDCG | MRR |
| *ours* | 0.773 | 0.755 | 0.854 | 0.027 | 0.809 | 0.807 | 0.826 | 0.825 | *ours* | 0.205 | 0.095 |
| w/o user | 0.757 | 0.731 | 0.841 | 0.033 | 0.791 | 0.790 | 0.798 | 0.796 | w/o text | 0.181 | 0.072 |
| w/o pre-training | 0.760 | 0.744 | 0.841 | 0.031 | 0.793 | 0.791 | 0.800 | 0.798 | w/o pre-training | 0.196 | 0.087 |

Table 2: Results of ablation studies on text-related tasks and user-related Tasks

## 4.2 Result Understanding

**Why community information help?** As Table 1 shows, graph-based models which utilize community insights perform better than language models on some datasets. Take NORMVIO as an example, NORMVIO is a dataset composed of comments which violate their community's norm and are deleted by moderators of community. In Reddit, the rules moderators enforce vary widely both in their formulation and interpretation across communities, making a one-size-fits-all approach increasingly brittle.

We analyze the subreddits that show improvement after introducing community information and find that samples misclassified by language models in these subreddits are usually due to violating community-specific norms which are different from common norms as shown in Figure 3(a). For example, the graph-based model has learned that hyperlinks are likely to be considered norm-violating in r/girlsinyogapants. r/CanadaPolitics has stricter civilization requirements, and r/changemyview has stricter formatting requirements.

**Why user nodes help?** User nodes can provide structural information by connecting two community nodes or two text nodes, and they are also associated with ground truth. By analyzing two classification text-related datasets NORMVIO and Dreaddit, we find that among users whose text quantity is greater than 1, the majority of their texts have only one text label. For example, as shown in Figure 3(b), in NORMVIO, there are 4541 users who have more than at least two comments in the dataset. 3300 of them have only one text label. Among the remaining 1241 users with two text labels, the proportion of the major label averaged by users is 56.05%. This proportion is shown in Figure 3(b) denoted by *.

**Why text nodes help?** Tabel 2 has shown the improvement brought by text nodes. Intuitively, we

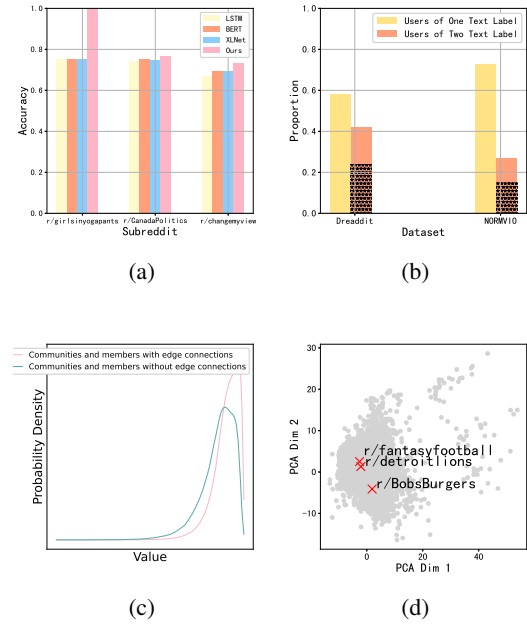

Figure 3: (a) Subreddits with improved results after adding community information (b) Statistics of users and their text labels (c) Probability density distribution of cosine similarity between users and their interested & not interested subreddits (d) Subreddit visualization

believe that active users have a closer semantic representation to the communities they are interested in. We get the semantic representation of community and user by averaging their texts' embedding produced by XLNet. Then we draw the probability density distribution of cosine similarity between users and their interested & not interested subreddits. As shown in Figure 3(c), active users have a closer semantic representation to the communities they are interested in. This can partially explain why text nodes work.

**Why pretrain help?** We analyze the changes in evaluation metrics of validation set during the first five epochs when incorporating pre-training and not incorporating pre-training. As shown in Figure 4, pre-training allows the model to obtain better ini-

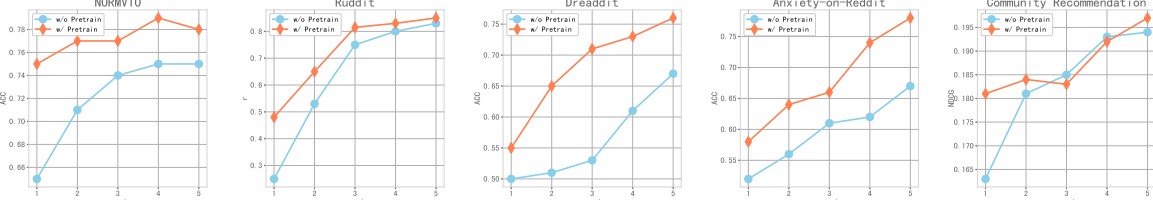

Figure 4: Improvement brought by the pre-trained model during the first 5 epochs on 5 datasets

tialization, leading to improved performance and faster convergence during the initial stages of training.

### 4.3 Community Visualization

We perform Principle Component Analysis (PCA) on subreddits' embedding generated in pre-training process(Figure 3(d)) . Back to the example in 1, we can intuitively see that r/fantasyfootball and r/BobsBurgers show higher cosine similarity than r/fantasyfootball and r/BobsBurgers.

## 5 Related Work

### 5.1 Pretrained Language Model in Social Network

In 2017, Google's team completely abandoned the traditional network structure, and only used the attention mechanism for machine translation tasks (Vaswani et al., 2017). In 2018, BERT performed well on many tasks with its self-supervised pre-training tasks and deep bidirectional models (Devlin et al., 2018). Subsequently, BERT and its variants have also been widely used in social network problems such as BERTweet (Nguyen et al., 2020), PoliBERTweet (Kawintiranon and Singh, 2022), UPPAM (Mou et al., 2023), and MentalBERT (Ji et al., 2021).

### 5.2 Methods in Community Profiling

Connection-based and content-based solutions are two mainstream ways of modeling online community. Connection-based solutions (Mishra et al., 2018; Arviv et al., 2021; Kumar et al., 2018; Martin, 2017; Janchevski and Gievska, 2019) are based on the idea that similar communities house similar users. The core focus of connection-based solutions is on users. Content-based solutions (Strzalkowski et al., 2020; Trujillo et al., 2021; Khalid et al., 2022) usually believe similar communities have similar semantics. The core focus

of connection-based solutions is on text in social media.

Our research uses both textual and user information, which allows for a more comprehensive and complete modeling of online community.

## 6 Conclusion

In this paper, we propose a unified graph pre-training model for online community modeling. It contains communities, users, and textual content simultaneously and can be easily used in various downstream tasks related to social media. Extensive experimental results validate the effectiveness and generalization capability of our approach.

## 7 Limitations

Our work aims to consider the various components and relationships within online communities as much as possible. The vast amount of data in online communities necessitates customized designed self-supervised tasks, especially focusing on reducing memory use and enhancing pre-training efficiency. Our model has a high memory usage due to the large volume of text and the use of pre-trained language models, leaving much room for further exploration.

## 8 Ethics Statement

### 8.1 Data Use

**Data Source Compliance** Data collection requires ethical reflection and precautions to preserve the dignity and privacy of users. Our experimental data comes from Pushshift. Pushshift is a Reddit data repository and now it is open for subreddit moderators who have Pushshift access. What's more, Pushshift is actively working with Reddit to find a solution to provide scholars with data access. Pushshift has provisions to delete data from the dataset upon user's request.

**Data Release** We release data in a manner that complies with the General Data Protection Regulation (GDPR). We don't provide any user-specific information. We release only the comment IDs and submission IDs. Reddit's Terms of Service do not prohibit the distribution of IDs. The researchers using the dataset need to retrieve the data with Pushshift access.

**Personal Privacy** We focus on protecting personal privacy throughout the entire research. Users are anonymous in data. Text-user edge generation used in pre-training does not lead to problems of individual identification. This task is achieved through negative sampling and contrastive learning, which cannot be directly used to identify individuals in reality. We won't release any user-specific information.

### 8.2   Societal Impacts

**Dual Use of Approaches** The correct use of our research will have a positive impact on social networks by solving social problems without any stance or commercial purpose. But we also accept that it may have adverse effects on society such as leaking personal data, violating personal privacy, threatening users' right to freedom of speech, and being abused to achieve personal interests, etc. Our proposed model will be shared selectively and subject to the approval of the Institutional Review Board (IRB), guaranteeing that any research or application based on this study is for research purposes only. Any attempt to use the proposed model to infer user characteristics and violate user privacy is strictly prohibited.

**Community Recommendation** Community recommendation (Janchevski and Gievska, 2019; Sundaresan et al., 2014) can quickly meet users' social needs and improve users' online experience. But it may indeed cause some potential social harm such as the leakage of personal data, the invasion of personal privacy by studying personal characteristics, and user disturbance caused by inaccurate or excessive recommendations. In order to mitigate the potential harm, the system can develop more fine-grained methods to learn the user's interest, such as using degree of belonging instead of 0/1 for recommendations. And it is necessary to fully respect the user's wishes. If the user does not want to be recommended, the system will not read any of his historical data.

### 8.3   Bias Alleviating

Text on social media can incorporate unintentional biases, so our model may inevitably be biased. To alleviate these problems, we will incorporate bias removal in future work to make our models fairer. In particular, mitigating bias can be approached from three aspects: cleaning data, adjusting models, and assessing the bias level in pre-trained models. Data cleaning can be done by filtering biased communities (Ferrer et al., 2021) and texts (Hube et al.). Previous research on adjusting the model focuses on how to involve bias reduction in the model structure (Xia et al., 2020; Mozafari et al., 2020). There have been some methods to evaluate the degree of bias of pre-trained models (Kurita et al., 2019; Field and Tsvetkov, 2019), and we can use them to evaluate our models.

### Acknowledgements

This work is supported by National Natural Science Foundation of China (No. 6217020551 and No. 62206056) and Science and Technology Commission of Shanghai Municipality Grant (No.21QA1400600).

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

# A  Appendix

## A.1  Pretrain Dataset Description

The data description of pre-training dataset is shown in Table 3

| Dataset | |
|---|---|
| # of total communities | 5126 |
| # of total users | 987945 |
| # of total texts | 1887178 |
| # of total community-user edges | 8701250 |
| # of total community-text edges | 1887178 |
| # of total user-text edges | 1887178 |

Table 3: Summary statistics of pre-training datasets

## A.2  Downstream Dataset

We crawl the data based on the IDs for published datasets. Because some data has been deleted by the original author, we usually cannot obtain a complete dataset. For unpublished datasets, we rebuild the dataset ourselves according to the data collection method in paper. For published data without validation set, we divide one-fifth of the training set into validation set. For the dataset we built ourselves, we divide the dataset into training set, validation set and test set as 70:15:15. The statistics of downstream dataset is shown in Table 4.

| | Target Objects | # of Target Objects |
|---|---|---|
| NORMVIO | Text | 44643 |
| Ruddit | Text | 5937 |
| Dreaddit | Text | 3517 |
| Anxiety-on-Reddit | Text | 30920 |
| Community Recommendation | User | 242840 |

Table 4: Summary statistics of downstream datasets

### A.3 Training Details

We provide our parameters in order to quickly apply the pre-trained model to downstream tasks. Before formal pre-training, we sample 10% of the data to help select appropriate parameters because pre-training on all data is price-expensive and time-consuming.

| Pre-training | |
|---|---|
| $\lambda$ | 0.5 |
| # of Sampled Negative Edges | 255 |
| Queue Size | 256 |
| Sample Depth | 6 |
| Sample Width | 128 |
| Hidden Dimension | 768 |
| Attention Head | 8 |
| GNN Layers | 3 |
| Dropout | 0.2 |
| Gradient Norm Clipping | 0.5 |
| Maximum Learning Rate | 1e-3 |
| Fine-tuning | |
| Sample Depth | 6 |
| Sample Width | 128 |
| Gradient Norm Clipping | 0.5 |
| Learning Rate | 5e-4 |
| Dropout | 0.2 |

Table 5: Training details