# OpenReview forum: "One-Model-Connects-All: A Unified Graph Pre-Training Model for Online Community Modeling"
_EMNLP/2023/Conference — EMNLP 2023 Findings_

### Official Review · Reviewer_uL1E · 2023-08-04

**Typos Grammar Style And Presentation Improvements:** 013, Attrbution -> Attribution

017, …
**Soundness:** 3

**Ethical Concerns:**

Yes

**Excitement:**

3: Ambivalent: It has merits (e.g., it reports state-of-the-art results, the idea is nice), but there are key weaknesses (e.g., it describes incremental work), and it can significantly benefit from another round of revision. However, I won't object to accepting it if my co-reviewers champion it.

**Justification For Ethical Concerns:**

As stated in Section 8.1, it may not be great to use Reddit data as the pretraining data, as pushshift is in violation of Reddit's Data API terms.

**Paper Topic And Main Contributions:**

This paper presents a unified pretraining framework involving communities, users, and textual content simultaneously. Graph-based approaches are employed to model the relationship between these three elements. Two self-supervised tasks including attribution generation and link generation are used to pre-train this ternary graph, making it easy to adapt to multiple downstream tasks related to social media. The pretraining data were constructed from Reddit and downstream task datasets are also Reddit-related. Experimental results show that pretraining can help achieve better results in 4 out of 5 downstream tasks.

**Questions For The Authors:**

1. The overall pretraining objective involves a hyperparameter \lambda. Considering that pretraining is costly, how do you determine the value of \lambda before starting pretraining?

2. "The entire pre-training procedure consumes a lot of memory, so it is conducted on only one GeForce RTX 3090 GPU", in this case, shouldn't multiple GPU cards be more helpful?



**Reasons To Accept:**

1. The paper is in general well-written and easy to follow.

2. Pretraining for online community modeling is of practical value and is worthy of study.

3. The proposed pretraining framework is simple yet effective, as verified by the downstream experiments.

**Reasons To Reject:**

1. The pretraining dataset is relatively small, it contains only 5K communities and less than 1M users. This work considers only the top 2% of subreddits that have a large number of posts. However, in practice, many subreddits would have only a few posts, these subreddits (or communities) should also be taken into account.

2. In the proposed framework, each text is treated as a node in the heterogeneous graph, which could lead to a large number of nodes, which limits the utilization of the framework. Besides, it is unclear why treating text is a good choice, are there other approaches to modeling text? For example, treating textual information as user node features.

3. In section 2.1.2, each relation (edge type) is represented by two matrices (A, B, C, D, E, F). Based on my understanding, the two matrices are the same. Can you explain why two matrices are used? It also seems that many matrices are not used in the following text.

4. The pretraining tasks only consider nodes corresponding to text content and edges that have text as one node. This ignores community nodes and user (author) nodes and the relationship between the community node and user node.

5. All five downstream datasets are related to Reddit. It is not sure whether the pre-trained model could generalize to datasets from other sources such as Twitter, Facebook.


**Reproducibility:**

3: Could reproduce the results with some difficulty. The settings of parameters are underspecified or subjectively determined; the training/evaluation data are not widely available.

**Reviewer Confidence:**

4: Quite sure. I tried to check the important points carefully. It's unlikely, though conceivable, that I missed something that should affect my ratings.

---

> ### Author Rebuttal · Authors · 2023-08-29
>
> 1. Take into account  subreddits which have only a few posts.
>
>     Response: Subreddits with few posts are rarely involved in Reddit related tasks, so we have not considered them.
>
> 2. Why use two matrices.
>
>     Response: These two matrices are transposed matrices, which are used to illustrate that there are two types of edges between two nodes. For example, between the user node and the text node, there are two kinds of edges: "user-write-text" and "text-from-user".
>
> 3. whether the pre-trained model could generalize to datasets from other sources such as Twitter, Facebook.
>
>     Response: The pre-trained model is pre-trained on the Reddit dataset, so it is difficult to adapt to other social platforms. But the whole framework of modeling online communities can be generalized to other social media. The only difference is that on social platforms such as twitter, there is no strict concept of community. So we need to artificially set the scope of the community. For example, consider a hashtag discussion topic as a community.
>
> 4. how to determine the value of  $\lambda$?
>
>     Response: Pre-training is price-expensive and time-consuming. Before formal pre-training, I sampled 10% of the data for pre-training to help me select appropriate parameters.
>
> 5. Conducted on only one GeForce RTX 3090 GPU.
>
>     Response: This is the shortcoming of my experiment, because the constructed graph is too large, and my pre-training method needs to read their initialization vectors into memory at one time, almost occupying the entire memory.

---

### Official Review · Reviewer_equG · 2023-08-05

**Soundness:** 3

**Excitement:**

3: Ambivalent: It has merits (e.g., it reports state-of-the-art results, the idea is nice), but there are key weaknesses (e.g., it describes incremental work), and it can significantly benefit from another round of revision. However, I won't object to accepting it if my co-reviewers champion it.

**Paper Topic And Main Contributions:**

This paper primarily focuses on the online community modeling area and proposes a unified graph pre-training model as the basis to apply to downstream tasks. The authors think previous research hasn’t fully utilized the information communities, users, and texts in social media. Therefore, they construct a ternary heterogeneous graph to comprehensively model the social network and propose a text generation task and the edge generation task to pre-trained an HGT network. The authors conducted extensive experiments on five Reddit-related datasets for hate speech detection and stress analysis tasks, demonstrating that the proposed model outperforms pre-trained language models such as BERT and XLNet, and some graph models.


**Questions For The Authors:**

If using XLNet to extract text representation and feed it to HGT to encode the structure of social networks, will do these operations have the same effect as pre-training a neural network?


**Reasons To Accept:**

(1) The paper has some nice ideas and some interesting insights that may be of benefit to the social network modeling community. And good writing makes this paper easy to follow.

(2) Better performance than baselines on five real-world datasets. The proposed framework achieves significant improvement on five datasets for two downstream tasks over several strong baselines, which shows the effectiveness of the framework.

(3) Extensive empirical analysis of the framework. The authors conduct a series of ablation studies to further prove the effectiveness of the framework.


**Reasons To Reject:**

(1) The novelty of this paper is a little limited. The basis models of this framework are HGT and XLNet, and the pre-trained tasks are node generation and edge generation, all of them are proposed by other research work. It seems this framework just stacks these models together.

(2) The fundamental reason for the improvement in model performance has not been clearly stated. It lacks experimental results to prove whether the improvement in performance comes from the design of a new framework or the integration of multiple features.


**Reproducibility:**

2: Would be hard pressed to reproduce the results. The contribution depends on data that are simply not available outside the author's institution or consortium; not enough details are provided.

**Reviewer Confidence:**

4: Quite sure. I tried to check the important points carefully. It's unlikely, though conceivable, that I missed something that should affect my ratings.

---

> ### Author Rebuttal · Authors · 2023-08-29
>
> 1. If using XLNet to extract text representation and feed it to HGT to encode the structure of social networks, will do these operations have the same effect as pre-training a neural network?
>
>     Response: Our baseline graph models use XLNet to extract text representations and feed them to graph neutral network(such as HGT)  to encode the structure of social networks. They are not as effective as our model. This indicates that pre-training a neural network is useful. What's more, we compare the experimental results with and without pre-training, as shown in Chapter 4.3(Why pretrain help). The only difference between the two curves is whether they have been pre-trained, and the other experimental steps are exactly the same.

---

### Official Review · Reviewer_2SWw · 2023-08-08

**Typos Grammar Style And Presentation Improvements:** Line 237, 260
**Soundness:** 2

**Excitement:**

2: Mediocre: This paper makes marginal contributions (vs non-contemporaneous work), so I would rather not see it in the conference.

**Paper Topic And Main Contributions:**

This paper proposes a pre-training framework for a graph of online community where there exist heterogeneous nodes types such as users, communities, and texts. It first constructs a graph from online communities data and initializes the node embeddings by using textual information. Then, the nodes are given to a GNN model to leverage the network structural information for better node representations. It uses two different self-supervised tasks, text reconstruction and edge generation for pre-training. In order to demonstrate the learned node representations are useful, the authors did experiments with downstream tasks and show the proposed method can achieve better performance across different datasets, compared to the baselines.

**Reasons To Accept:**

* Given the complexity and size of the online community data like Reddit, the proposed is able to effectively learn the representations of different types of entities.
* The experiments were done with various baselines/tasks along with ablation study, which cover various aspects of the community modeling.
* Overall the paper is easy to read and follow although it needs to improve presentation a little bit.

**Reasons To Reject:**

* Not sure how well the proposed method is applicable to other datasets/problems. It seems to heavily rely on the textual information, and thus the proposed method might not be effective for some online communities in which enough amount of texts is not available.
* Lack of technical novelties. How to initialize the nodes and the objectives used to optimize them have been commonly used in previous works.
* Although three different types of nodes are considered, they are eventually initialized based on texts. So, they might eventually make contributions in the same way by adding more textual information, rather than they actually provide structural/community information. There needs other experiments to verify it.
* There is no connection between the same type nodes, and

**Reproducibility:**

3: Could reproduce the results with some difficulty. The settings of parameters are underspecified or subjectively determined; the training/evaluation data are not widely available.

**Reviewer Confidence:**

4: Quite sure. I tried to check the important points carefully. It's unlikely, though conceivable, that I missed something that should affect my ratings.

---

> ### Author Rebuttal · Authors · 2023-08-29
>
> 1. Not sure how well the proposed method is applicable to other datasets/problems.
>
>     Response: This paper focuses on the online community in Reddit and selects five datasets from Reddit as downstream tasks.  These five datasets are not only broad in type, as they can be divided into node-related&edge-related and text-related&user-related, but also rich in content, including popular tasks on Reddit in recent years. So we believe that these datasets are representative, and the experimental results on these datasets can represent the effectiveness of our model in modeling the Reddit online community.
>
> 2. Not effective for some online communities in which enough amount of texts is not available.
>
>     Response: For a community without a lot of text information, although it cannot obtain a good representation in the initialization stage, the model will provide additional information through communities with the same users during the training process, which is very friendly for communities with few texts.
>
> 3. Although three different types of nodes are considered, they are eventually initialized based on texts. So, they might eventually make contributions in the same way by adding more textual information, rather than they actually provide structural/community information. There needs other experiments to verify it.
>
>     Response: Although all three types of nodes are initialized through text, they not only contribute by supplementing additional text information. During pre-training, predicting the structure is a pre-training task which can enable the pre-trained model to learn structural information from graph. So the pre-trained model contains structural information.
>
>     Our baseline pre-trained language models have been further pre-trained on the collected Reddit historical data, with epoch set to 2. And they are not as effective as our model. This indicates that structural information is useful in pre-trained models.

---

### Official Review · Reviewer_bxaU · 2023-08-09

**Typos Grammar Style And Presentation Improvements:** 1. line 074 `facilities`  -> `facilit…
**Soundness:** 3

**Excitement:**

3: Ambivalent: It has merits (e.g., it reports state-of-the-art results, the idea is nice), but there are key weaknesses (e.g., it describes incremental work), and it can significantly benefit from another round of revision. However, I won't object to accepting it if my co-reviewers champion it.

**Missing References:**

1. Veličković, P., Fedus, W., Hamilton, W. L., Liò, P., Bengio, Y., & Hjelm, R. D. (2018). Deep graph infomax. arXiv preprint arXiv:1809.10341.

**Paper Topic And Main Contributions:**

This paper proposes a pre-training framework to leverage the structural strength of heterogeneous graphs and the semantic strength of textual content (in the context of Reddit) to enable downstream tasks for community-related analysis. Primary contributions as per the author are (1) the proposed pre-training framework, (2) the graph-based approach, and (3) an analysis of downstream tasks on social media.

**Questions For The Authors:**

A. Have you considered how Deep Graph Infomax for graph-based pre-training differs from your framework? I believe some comparisons there would be useful in showing the theoretical benefits of your work, particularly since the baselines are not pre-trained either.
B. Did the authors finetune the baselines directly on the downstream tasks? Or did they pre-train the baselines on their pre-training dataset and then fine-tune them to get comparable results?

**Reasons To Accept:**

1. The work has great potential application in merging both the structural (community user graphs) and semantic (text content) information for a social network analysis - particularly in computational social science.
2. A good analysis of downstream tasks, and a strong improvement upon the author's chosen baselines.
3. The discussion regarding why each part of the model (text node, user node) is helpful which is a strong point of this paper.

**Reasons To Reject:**

1. In line 093 the authors identify that "extensive experiments have to be completed to confirm the effectiveness of our frameworks" indicating that it may be premature to make the claims about the benefit of this paper.
2. The primary research question and the problem the authors are trying to address is unclear in the paper. They mention in the abstract that their study will "provide assistance on how to better model online community", but they are unclear exactly how.
3. Another weakness of this paper is in its experimental setup and experiments. The chosen baselines are justified, however, I am unsure about the process in which said baselines were used. Did they finetune the baselines directly on the downstream tasks? Or did they pre-train the baselines on their pre-training dataset and then fine-tune to get comparable results? I am including this weakness as a question to the authors.
4. Deep Graph Infomax (DGI) exists as a well-established graph-based pre-training approach (node representation tasks only) - the pre-training framework suggested by the author while well thought out and developed would be much better established in comparison to DGI. (Included in missing references).

**Reproducibility:**

4: Could mostly reproduce the results, but there may be some variation because of sample variance or minor variations in their interpretation of the protocol or method.

**Reviewer Confidence:**

4: Quite sure. I tried to check the important points carefully. It's unlikely, though conceivable, that I missed something that should affect my ratings.

---

> ### Author Rebuttal · Authors · 2023-08-29
>
> 1. It may be premature to make the claims about the benefit of this paper.
>
>     Response: This paper focuses on the online community in Reddit and selects five datasets from Reddit as downstream tasks.  These five datasets are not only broad in type, as they can be divided into node-related&edge-related and text-related&user-related, but also rich in content, including popular tasks on Reddit in recent years. So we believe that these datasets are representative, and the experimental results on these datasets can represent the effectiveness of our model in modeling the Reddit online community.
>
> 2. The primary research question and the problem the authors are trying to address is unclear in the paper.
>
>     Response: The main problem that this paper wants to address is how to better model online communities, and then use the communities to provide assistance for social media related tasks. Relevant studies have proved that using community information can assist tasks such as abusement detection[1] and conflict analysis[2]. This article aims to provide help on how to make better use of communities. Specifically, we consider community, user, and text simultaneously, instead of only considering community and user[3-5] or community and text[6-7], and we use graph structure to simulate the relationship between these three items, instead of simply adding community's name[1] after text or concatenated community's vectors after text's vectors. Finally, we also use the rich historical data on social media to pretrain a pre-trained model, so that it can be more easily to adapt to multiple downstream tasks.
>
> 3. About baselines.
>
>    Response: We have continued pre-training the baseline pre-trained language models on the collected historical Reddit data, and the epoch is set to 2.
>
> 4. About DGI.
>
>    Response: DGI relies on maximizing mutual information between patch representations and corresponding high-level summaries of graphs—both derived using established graph convolutional network architectures.  It is not suitable for our task, mainly for the following reasons.
>
>    (1) The graph we constructed is heterogeneous, and DGI does not have any additional advantages at the heterogeneous graph level.
>
>    (2) The construction of DGI's negative samples and global information has a significant impact on the results.
>
>    (3) In social networks, we believe that "close" nodes should also be more "close" in the representation space, as they usually have similar features in reality. So random walk is suitable.
>
> References:
>
> [1] Chan Young Park, Julia Mendelsohn, Karthik Radhakrishnan, Kinjal Jain, Tushar Kanakagiri, David Jurgens, and Yulia Tsvetkov. 2021. Detecting community sensitive norm violations in online conversations. arXiv preprint arXiv:2110.04419.
>
> [2] Marilia A Calcia, David R Bonsall, Peter S Bloomfield, Sudhakar Selvaraj, Tatiana Barichello, and Oliver D Howes. 2016. Stress and neuroinflammation: a systematic review of the effects of stress on microglia and the implications for mental illness. Psychopharmacology, 233:1637–1650.
>
> [3] Srijan Kumar, William L Hamilton, Jure Leskovec, and Dan Jurafsky. 2018. Community interaction and conlict on the web. In Proceedings of the 2018 world wide web conference, pages 933–943.
>
> [4] Trevor Martin. 2017. community2vec: Vector representations of online communities encode semantic relationships. In Proceedings of the Second Workshop on NLP and Computational Social Science, pages 27–31.
>
> [5] Isaac Waller and Ashton Anderson. 2019. Generalists and specialists: Using community embeddings to quantify activity diversity in online platforms. In The World Wide Web Conference, pages 1954–1964.
>
> [6] Tomek Strzalkowski, Anna Newheiser, Nathan Kemper, Ning Sa, Bharvee Acharya, and Gregorios Katsios. 2020. Generating ethnographic models from communities’ online data. In Proceedings of the Second Workshop on Figurative Language Processing, pages 165–175.
>
> [7] Milo Z Trujillo, Samuel F Rosenblatt, Guillermo de Anda Jáuregui, Emily Moog, Briane Paul V Samson, Laurent Hébert-Dufresne, and Allison M Roth. 2021. When the echo chamber shatters: Examining the use of community-specific language post-subreddit ban. arXiv preprint arXiv:2106.16207.

---

### Meta-Review · Area_Chair_5RK1 · 2023-09-19

**Recommendation:** 3

**Metareview:**

The paper investigates a unified graph pre-training method for online communities. The reviewers appreciated the novelty, method, experiments, and analysis. The authors are recommended to address the reviewers' concerns in the camera ready, if the paper is accepted.

---

### Decision · Program_Chairs · 2023-10-07

**Decision:**

Accept-Findings

**Comment:**

The paper investigates a unified graph pre-training method for online communities. The reviewers appreciated the novelty, method, experiments, and analysis. The authors are recommended to address the reviewers' concerns in the camera ready, if the paper is accepted.